# Etiology and Oral Antibiotic Susceptibility Patterns of the First Urinary Tract Infection Episode in Infants Under 6 Months of Age: A 17-Year, Retrospective, Single-Center Study in Italy

**DOI:** 10.3390/microorganisms13030607

**Published:** 2025-03-06

**Authors:** Francesca Bagnasco, Francesca Lorenzini Ceradelli, Alessio Mesini, Carolina Saffioti, Erica Ricci, Chiara Russo, Marcello Mariani, Elisabetta Ugolotti, Emanuela Caci, Debora Contu, Elisa Tavella, Alessia Cafaro, Giorgio Piaggio, Enrico Verrina, Roberto Bandettini, Elio Castagnola

**Affiliations:** 1Epidemiology and Biostatistics, Scientific Directorate, IRCCS Istituto Giannina Gaslini, 16147 Genova, Italy; 2Department of Neuroscience, Rehabilitation, Ophthalmology, Genetics and Maternal and Child Sciences (DiNOGMI), University of Genoa, 16132 Genoa, Italy; francescalorenziniceradelli@gaslini.org (F.L.C.); chiararusso@gaslini.org (C.R.); 3Infectious Disease Unit, Department of Pediatrics, IRCCS Istituto Giannina Gaslini, 16147 Genova, Italy; alessiomesini@gaslini.org (A.M.); carolinasaffioti@gaslini.org (C.S.); ericaricci@gaslini.org (E.R.); marcellomariani@gaslini.org (M.M.); eliocastagnola@gaslini.org (E.C.); 4Laboratory of Clinical Analysis, Department of Pediatrics, IRCCS Istituto Giannina Gaslini, 16147 Genova, Italy; elisabettaugolotti@gaslini.org (E.U.); emanuelacaci@gaslini.org (E.C.); deboracontu@gaslini.org (D.C.); elisatavella@gaslini.org (E.T.); alessiacafaro@gaslini.org (A.C.); 5Nephrology and Kidney Transplant Unit, Department of Pediatrics, IRCCS Istituto Giannina Gaslini, 16147 Genova, Italy; giorgiopiaggio@gaslini.org (G.P.); enricoverrina@gaslini.org (E.V.); robertobandettini@gaslini.org (R.B.)

**Keywords:** urinary tract infections, infants, oral antibiotic susceptibility, etiology, Italy

## Abstract

Background: Urinary tract infections (UTIs) are among the most common bacterial infections in children, and the antibiotic susceptibility in the youngest patients remains poorly understood. This study aimed to describe the distribution of uropathogens and their antibiotic susceptibility, focusing on oral formulations. Methods: Data from the first microbiological isolation, between January 2007 and December 2023, at Istituto Gaslini, in young infants (aged <6 months), were analyzed. Results: We isolated 2473 infants’ first pathogen, with a median age in the sample of 2.8 months and 62.6% male. A total of 2498 bacterial isolates were identified, of which 88.8% were Gram-negative and 11.2% were Gram-positive. *Escherichia coli* (53%) was the most frequent isolate, followed by *Klebsiella pneumoniae* (12.3%) and *Enterococcus* spp. (9.6%). No significant differences were observed between males and females, but infants younger than 3 months exhibited a significantly different pathogen distribution compared to older infants. The pathogen distribution showed significant changes before and after 2015, with a marked increase in *Klebsiella pneumoniae* isolates post-2015. *Escherichia coli* showed increases in resistance to amoxicillin-clavulanate and ciprofloxacin after 2015. Conclusions: *Escherichia coli* remains the most common uropathogen; however, *Klebsiella pneumoniae* has not only shown a high prevalence but also significant resistance, particularly in recent years.

## 1. Introduction

Urinary tract infections (UTIs) are among the most prevalent bacterial infections in pediatrics, with infants in the first few months of life being at a particularly high risk [1]. This susceptibility has been attributed to an incompletely developed adaptive immune system, but congenital malformations of the urinary tract also play a pivotal role, especially in males [2,3,4].

Gram-negative bacteria from the *Enterobacterales* family are the predominant cause, with *Escherichia coli* accounting for over 70% of UTIs. Other common pathogens include *Klebsiella* spp., *Enterobacter* spp., and *Proteus* spp., while *Pseudomonas aeruginosa* is less frequent but associated with severe infections [5]. Among Gram-positive bacteria, *Enterococcus* spp. are the most prevalent UTI pathogens. However, there is significant geographical variability in the prevalence of microorganisms and the patterns of antimicrobial resistance [6].

The increasing prevalence of antimicrobial resistance also complicates the management of UTIs in the pediatric population, with resistance to orally administrable first-line antibiotics such as amoxicillin-clavulanate documented worldwide. Recent reports from different countries have highlighted a concerning prevalence of extended-spectrum beta-lactamase (ESBL)-producing and multidrug-resistant strains of *Escherichia coli* and *Klebsiella* in children, underscoring the need for updated local data to guide empirical treatment approaches [5,7,8,9,10,11,12,13,14]. However, the antibiotic susceptibility of bacteria isolated from the urinary tract in the youngest subjects remains poorly explored, particularly for those suitable for oral administration.

It is well-documented that both the prevalence of microorganisms and antibiotic susceptibility exhibit geographical variability [6,15]. Such variability can manifest on a local scale over time, and these temporal changes can be effectively monitored through longitudinal studies. Accordingly, this 17-year retrospective study aimed to describe and analyze the etiology and antibiotic susceptibility patterns associated with the first bacterial isolation from urine cultures in young infants aged 0–6 months, observed at a pediatric tertiary care center in northwest Italy.

## 2. Materials and Methods

Data on microbiological isolation from urine cultures sampled between January 2007 and December 2023 in young infants, aged 0–6 months, were extracted from the Microbiology Laboratory database at the IRCCS Istituto Giannina Gaslini (IGG), Genoa, Italy and analyzed anonymously. The IGG is a tertiary care pediatric hospital in northwest Italy, admitting patients from Italy and other countries. For each episode, data on sex, age in months at the time of the first isolation, and the department of admission, where the patient was sampled, were retrieved.

All positive urinary cultures were considered, independently of a diagnosis of upper (pyelonephritis) or lower (cystitis) UTI or asymptomatic bacteriuria [8,16]. All urine samples were obtained by midstream clean-catch, catheterization, or urine bags, as determined by the patient’s age, according to international recommendations [17,18]. Urine samples were cultured on Columbia agar with 5% sheep blood (bioMérieux SA, Marcy-l’Etoile, France) and MacConkey agar (bioMérieux SA, Marcy-l’Etoile, France), then incubated at 37 °C overnight. Bacterial growth was considered significant if it reached ≥105 colony-forming units (CFU)/mL of urine. Antibiotic susceptibility testing was performed using automated systems (BD Phoenix, Becton, Dickinson and Company, Sparks, MD, USA), following the Clinical and Laboratory Standards Institute (CLSI) guidelines from 2007 to 2010. After 2011, the European Committee on Antimicrobial Susceptibility Testing (EUCAST) guidelines were used. Due to this major shift and annual updates to EUCAST breakpoints, we chose to report antibiotic susceptibility or resistance based on the criteria set by the system used each year, reflecting the protocols followed in routine practice over time.

The isolated pathogens were divided into Gram-positive and Gram-negative bacteria. Within the Gram-positive group, pathogens were further categorized as *Enterococcus* spp., *Streptococcus* spp., and *Staphylococcus* spp. In the Gram-negative group, pathogens were classified as *Enterobacterales* and glucose non-fermenting bacteria. Antibiotic susceptibility was analyzed for bacterial pathogens accounting for >3% of all isolates. For the Gram-negative strains, antibiotic susceptibility was assessed against amoxicillin-clavulanate, cefixime, cefuroxime, ciprofloxacin, fosfomycin, nitrofurantoin, cotrimoxazole, cefotaxime, and ceftazidime. In the case of *Enterococcus* spp., susceptibility was evaluated for nitrofurantoin and ampicillin. It is important to note that amoxicillin shares the same spectrum of activity as ampicillin and can therefore be used for oral therapy of ampicillin-sensitive *Enterococcus* strains. Due to differences in the antibiotics available in different periods for automated systems, nitrofurantoin and cefixime were tested from 2011 to 2015, cefuroxime from 2007 to 2015, and cotrimoxazole from 2007 to 2021.

### Statistical Analysis

Descriptive statistics were reported in terms of absolute frequencies and percentages for categorical data, and Pearson’s chi-square test or Fisher’s exact test, when appropriate, was applied to compare proportions. Continuous data were described in terms of median values and IQRs, due to their non-normal (Gaussian) distribution.

The year and age at the first isolation were categorized based on the median values of 2015 and 2.8 months, respectively, with the latter being approximated to 3 months. Percentages of antibiotic-resistant pathogen were calculated as the ratio of the number of resistant to the number of tested strains. The analysis and presentation were based on the available data, i.e., no imputation of missing data was performed.

All tests were 2-tailed and a *p* value < 0.05 was considered statistically significant. All analyses were performed using Stata (Stata Corp. Stata Statistical Software, Release 18.0, College Station, TX, USA, Stata Corporation, 2023).

This study was conducted in accordance with the Helsinki Declaration. According to Italian legislation, this study did not need ethical approval, as it was a purely observational, retrospective study on routinely collected anonymous data. Moreover, informed consent for participation in this study was not required since retrospective data were obtained by an anonymous microbiology database. In any case, consent for the completely anonymous use of one’s clinical data for research/epidemiological purposes is requested as part of the clinical routine at the time of admission/during the diagnostic procedure.

## 3. Results

During the study period, 2473 young infants had their first isolation of pathogens from urinary samples at a median age of 2.8 months (IQR 1.2–4.7). The majority (62.6%) were male, and 25% of them subsequently had other episodes (Table 1). For more than half of the cases (55.2%), the department of first sampling was the Emergency Unit.

A total of 2473 positive cultures yielded 2498 bacterial isolates; 281 (11.2%) were Gram-positives, and 2217 (88.8%) were Gram-negatives (Table 2). Among the 2498 isolates, the most frequent pathogen was *Escherichia coli* (53%), followed by *Klebsiella pneumoniae* (12.3%) and *Enterococcus* spp. (9.6%). In the Gram-positives, the most frequent were enterococci (241, 85.8%), with *E. faecalis* being the most common species (n = 209). Among the Gram-negatives, 2116 (95.4%) were *Enterobacterales*, with *Escherichia coli* being the most frequent (n = 1324, 62.6%). Non- glucose fermenting bacteria totaled 101 (4.6%), with the majority (n = 90) being *Pseudomonas aeruginosa*.

### 3.1. Distribution of the Most Frequently Isolated Bacteria

The distribution of the most frequently isolated bacteria (n = 2201), including *Escherichia coli*, *Klebsiella pneumoniae*, *Klebsiella oxytoca*, *Enterobacter cloacae*, *Enterococcus faecalis*, and *Pseudomonas aeruginosa*, was not statistically significant different between males and females (Table 3). However, statistically significant differences (*p* < 0.001) were observed between infants younger than 3 months and those aged 3 months or older (Table 3).

The temporal trends (Figure 1) highlighted *Escherichia coli* as the dominant species; its frequency showed a sharp increase, starting in 2008, peaking in 2015, and then gradually declining. *Klebsiella pneumoniae* emerged as the second most prevalent species, displaying a steady rise in frequency from 2010 onward. If the median year (2015) of the study period was considered (Table 3), the pathogen distribution displayed statistically significant patterns (*p* < 0.001); *Klebsiella oxytoca*, *Enterobacter cloacae*, *Pseudomonas aeruginosa*, and *Enterococcus faecalis* were most frequently isolated before 2015, while there was almost an equal distribution for *Escherichia coli* in the two periods. After 2015, there was a higher frequency of *Klebsiella pneumoniae*.

### 3.2. Temporal Trends in Antibiotic Resistance by Pathogen

The distribution of antibiotic resistance overall, and by period (<2015 or ≥2015) of the first isolate, is summarized in Table 4**.** Annual trends in antibiotic resistance are shown in Figure 2. In *Escherichia coli*, the highest resistance was observed for amoxicillin-clavulanate (27.9%), followed by cotrimoxazole, with resistance trends exhibiting variability over the study period. Starting from 2015, resistance to amoxicillin-clavulanate and ciprofloxacin increased significantly (from 24.1% to 31.8%, *p* = 0.002; and from 5% to 10.7%, <0.001, respectively, Table 4) compared to the previous period. Both nitrofurantoin and fosfomycin showed minimal resistance. In *Klebsiella pneumoniae*, frequent resistance was observed to nitrofurantoin, amoxicillin-clavulanate, and third-generation cephalosporins. Resistance significantly increased from 2015 compared to the prior period for ciprofloxacin (from 2.7% to 12.8%), cotrimoxazole (from 12.3% to 23.3%), cefotaxime (from 11.9% to 22.4%), and ceftazidime (from 13.7% to 23.1%). *Klebsiella oxytoca* showed infrequent resistance to ciprofloxacin, cotrimoxazole, and ceftazidime, while maintaining high resistance to cefuroxime, during the period in which it was tested, and exhibited fluctuating resistance to amoxicillin-clavulanate, with notable peaks throughout the study period. *Enterobacter cloacae* displayed high resistance to amoxicillin-clavulanate and substantial resistance to cefixime and cefuroxime. Ciprofloxacin exhibited minimal resistance. Although *Pseudomonas aeruginosa* was less frequently isolated, it showed a variation in resistance over time, with peaks especially for ceftazidime compared to ciprofloxacin. *Enterococcus faecalis* exhibited excellent susceptibility to ampicillin and nitrofurantoin, which followed similar temporal trends.

## 4. Discussion

This study investigates the etiology and antibiotic resistance profiles of the first bacterial isolates from UTIs in young infants, under six months of age. We specifically focused on this age group because, in our previous study [8], we observed a concerning prevalence of resistance among infants. As the first single-center study to specifically examine bacterial profiles and antibiotic resistance in this population, it offers important insights into the evolving patterns of UTI pathogens and the growing challenge of antibiotic resistance. However, this study primarily addresses the epidemiology of bacterial isolates, independent of symptom presence to confirm UTIs, due to the lack of clinical data. While this constitutes a notable limitation, it still provides valuable information regarding antibiotic efficacy.

Another important consideration is the shift in the methodology used for antibiotic susceptibility testing in our center. From 2007 to 2010, testing was conducted according to the CLSI guidelines. However, starting in 2011, the EUCAST guidelines were adopted. This transition to EUCAST, which publishes annual updates to breakpoint values, required that antibiotic susceptibility be reported in accordance with the criteria defined by the guidelines in effect for each respective year. For instance, recently, EUCAST revised the breakpoints for amoxicillin-clavulanate, particularly for *Enterobacterales* associated with UTIs, highlighting the importance of adhering to the latest guidelines for accurate susceptibility reporting.

In line with previous findings [2,3,4,19,20], this study observed a higher prevalence of isolates in males. This is consistent with anatomical and physiological factors that predispose male infants to UTIs, while females typically show a higher incidence of UTIs starting from the second year of life, due to hormonal and anatomical changes. However, in this study, the distribution of the most commonly isolated pathogens did not show statistically significant differences between males and females. The present study also highlighted that the first bacterial isolation typically occurred within the first few months of life, with a median age of approximately 3 months. This early onset, combined with significant levels of antibiotic resistance, might be at least partially attributed to the acquisition of nosocomial microorganisms during birth, as previously reported in the literature [21,22,23]. Indeed, the significant differences in pathogen distribution between infants younger than 3 months and those older than 3 months suggest that microbial exposure and resistance acquisition may differ as infants mature. In fact, as another study [24] showed, positive urine cultures were most common in infants with a postnatal age of 8–30 days and a very low birth weight (<1500 g).

Our data confirm that UTIs in young infants are predominantly diagnosed in emergency settings. Clear criteria are needed to assist clinicians in deciding whether to test a patient’s urine and how to manage the case. Due to the necessary delay in obtaining urine culture results, clinicians must make decisions on whether to prescribe antibiotics for a suspected UTI prior to receiving culture results [25].

A further important observation is that recurrent UTIs were observed in 25% of young infants after their first isolation, which is consistent with other studies [26,27]. Recurrent infections are more common in the presence of urinary tract abnormalities [28], particularly those with vesicoureteral reflux, which often requires repeated antibiotic courses. This repeated exposure to antibiotics is a well-established risk factor for the development of antibiotic resistance. As already mentioned, a limitation of this study is the lack of detailed clinical data, such as data on the presence of urinary tract malformations, which would help better contextualize these findings.

Gram-negative bacteria were the predominant pathogens, with *Escherichia coli* being the most commonly isolated pathogen, though at a lower frequency (53%, among all isolates) compared to the >70% reported in the literature. The second most frequently isolated pathogen was *Klebsiella pneumoniae* (12.3%, among all isolates). However, there is considerable variability in the prevalence of pathogens, driven both by geographical differences and by the specific study populations and time periods. For instance, a single-center study conducted in Portugal [29] between 2017 and 2019, which included patients up to 18 years of age, found that *Escherichia coli* was the most commonly isolated microorganism (71.5%), followed by *Proteus mirabilis* (14.9%) and *Klebsiella pneumoniae* (5.1%). A study of a specific multicenter cohort in the United States [24] involving infants with very low birth weight, who were discharged from the neonatal intensive care unit, found that *Enterococcus* spp. (20%) was the most common pathogen, followed by *Escherichia coli* (19%) and *Klebsiella* spp. (18%).

The data further revealed a notable shift in pathogen distribution over time. A greater number of pathogens were isolated prior to 2015, particularly *Klebsiella oxytoca* and *Pseudomonas aeruginosa*, while *Escherichia coli* was almost equally distributed across the time period. After 2015, an increased frequency of *Klebsiella pneumoniae* was observed. This temporal shift in pathogen profiles highlights the importance of ongoing surveillance to understand local epidemiology and the necessity of adjusting empirical antibiotic treatment regimens accordingly. In fact, the selection of empirical antibiotics for pediatric UTIs is strongly influenced by local bacterial profiles and resistance patterns within specific populations [23]. Accurate knowledge of the responsible pathogens and their antibiotic susceptibility is essential, particularly in light of the dynamic nature of UTI etiology and the evolving resistance patterns of uropathogens. This is even more critical in very young infants, who are particularly vulnerable to severe infections and present unique bacterial profiles compared to older children or adults.

Our data on temporal trends in antibiotic resistance revealed that certain antibiotics (nitrofurantoin, cefixime, cefuroxime, and cotrimoxazole) were tested intermittently, limiting the generalizability of the findings across the entire study period. Amoxicillin-clavulanate and ciprofloxacin were the most commonly tested oral antibiotics, while ceftazidime was primarily tested for intravenous administration. High rates of antibiotic resistance were documented for commonly prescribed oral antibiotics in infants with their first bacterial isolates. This may be linked to pre- or peri-partum maternal antibiotic exposure, which can increase the risk of colonization by resistant bacteria in neonates [28]. Maternal–child transmission through colonization of the fecal microbiota or genital tract may represent an underrecognized risk factor for acquiring ESBL-producing *Enterobacterales* [27]. The observed high rates of antibiotic resistance, particularly to commonly used oral antibiotics like amoxicillin-clavulanate, raise concerns about the feasibility of adhering to national treatment guidelines, which recommend amoxicillin-clavulanate or cefixime as first-line therapies for pediatric UTIs [30]. The increasing resistance to amoxicillin-clavulanate, especially in *Escherichia coli* and *Klebsiella pneumoniae*, underscores the need for more cautious use of these antibiotics, particularly in neonates and infants under one year of age [23]. Ciprofloxacin consistently exhibited the lowest resistance rates across most pathogens. However, for *Escherichia coli* and *Klebsiella pneumoniae*, resistance significantly increased from 2015 onwards compared to the previous period. Notably, *Escherichia coli* showed a significant rise in resistance to amoxicillin-clavulanate, from 24% to 32%. These findings are consistent with a previous study in which *E. coli* strains demonstrated an increase in resistance to amoxicillin-clavulanate over time, from 16% (2000–2004) to 36% (2015–2019) [31]. In our earlier study [8], male infants aged <6 months, admitted to the Neonatal or Pediatric ICU with a history of previous infections, were found to have an estimated risk of over 60% for resistance to amoxicillin-clavulanate for *Escherichia coli* and other *Enterobacteriales*.

*Klebsiella pneumoniae* was most frequently isolated since 2015, particularly in young infants under 3 months of age, with an increase in resistance during the more recent period, both to oral and intravenous antibiotics. In a study in Poland [32], non-*E. coli* UTIs were significantly more common in patients with congenital anomalies of the kidney and urinary tract, neurogenic bladder, or those receiving antibiotic prophylaxis. Infections caused by non-*E. coli* and multidrug-resistant bacteria may be associated with prior hospitalizations and patient colonization by pathogens acquired in the hospital environment. Specifically, Devrim et al. [33] reported that nosocomial UTIs were most often caused by *Klebsiella pneumoniae* (34.1%).

## 5. Limitations

This study had several limitations, including its retrospective design and the fact that it was conducted within a specific local epidemiological context, limiting the generalizability of the findings. Additionally, certain antibiotics were tested only intermittently, which restricts the applicability of the results across the entire study period. Furthermore, the data for this study were extracted from a microbiology database, and as such, detailed clinical data were not available for analysis.

## 6. Conclusions

This study provides a comprehensive analysis of bacterial pathogens and their susceptibility profiles in young infants, under 6 months of age, over a 17-year period. *Escherichia coli* remains the most common uropathogen, with a notable increase in amoxicillin-clavulanate resistance in recent years. However, it is important to highlight that *Klebsiella pneumoniae* has not only shown a high prevalence but also significant resistance, particularly in recent years, to both oral and intravenous antibiotics. These findings underscore the need for more personalized and targeted antibiotic therapies, contributing to the optimization of empirical antibiotic treatment and supporting antimicrobial stewardship initiatives. Ultimately, this may improve clinical outcomes in this vulnerable population, reduce the time required for managing UTIs, and help minimize healthcare workloads and associated costs.

## Figures and Tables

**Figure 1 microorganisms-13-00607-f001:**
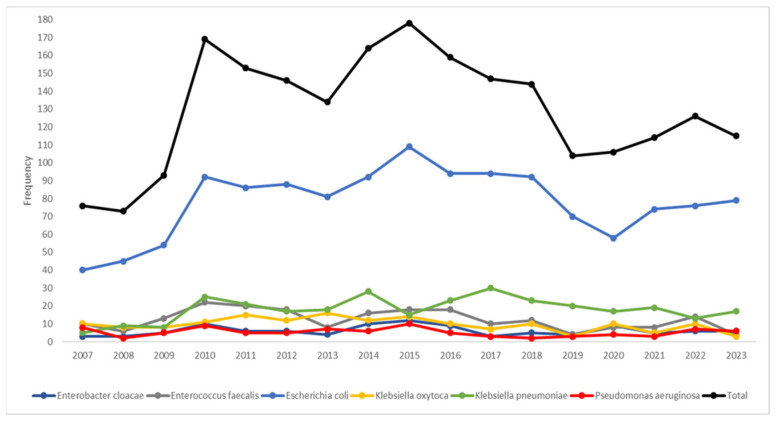
Temporal trends of the most frequent bacterial pathogens isolated (n = 2201) during the study period.

**Figure 2 microorganisms-13-00607-f002:**
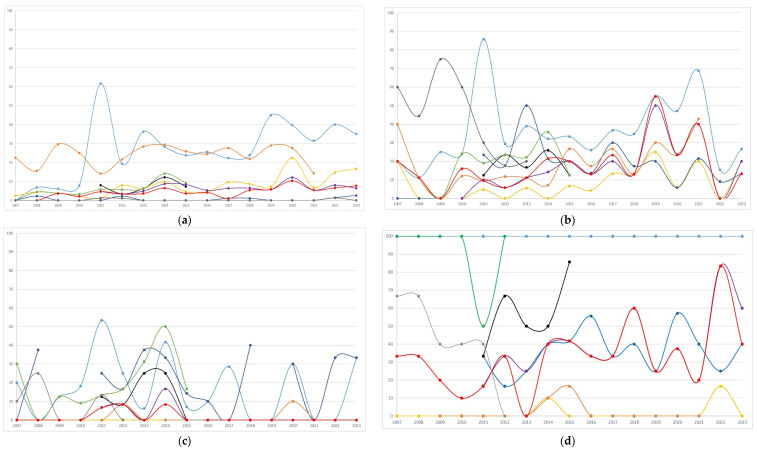
Temporal trends (percentages of resistant/tested) of antibiotic resistance to the most frequent bacterial pathogens isolated (n = 2201) during the study period: (**a**) *Escherichia coli*, n = 1324; (**b**) *Klebsiella pneumoniae*, n = 308; (**c**) *Klebsiella oxytoca*, n = 164; (**d**) *Enterobacter cloacae,* n = 106; (**e**) *Pseudomonas aeruginosa,* n = 90; (**f**) *Enterococcus faecalis,* n = 209.

**Table 1 microorganisms-13-00607-t001:** Characteristics of young infants, aged 0–6 months, at the first isolation of pathogens from urinary samples.

Total Infants, n (%)	2473 (100.0)
Sex, M	1549 (62.6)
Months at the 1st UTI, median (IQR)	2.8 (1.2–4.7)
Total of UTI, median (IQR), min–max values	1 (1–2), 1–23
1	1854 (75)
2	347 (14)
>2	272 (11)
Department of admission	
Emergency	1366 (55.2)
Neonatal/Pediatric ICU	336 (13.6)
Infectious Diseases	221 (8.9)
Surgery/Orthopedics/Neurosurgery	128 (5.2)
Nephrology	115 (4.7)
Hematology/Oncology	7 (0.3)
Others	300 (12.1)
Year of the 1st UTI, median (IQR)	2015 (2011–2018)
2007	99 (4.0)
2008	85 (3.4)
2009	118 (4.8)
2010	181 (7.3)
2011	181 (7.3)
2012	163 (6.6)
2013	151 (6.1)
2014	191 (7.7)
2015	201 (8.1)
2016	176 (7.1)
2017	163 (6.6)
2018	159 (6.4)
2019	114 (4.6)
2020	110 (4.4)
2021	125 (5.0)
2022	134 (5.4)
2023	122 (4.9)

**Table 2 microorganisms-13-00607-t002:** Distribution of 2498 bacterial pathogens isolated in 2473 UTIs.

	Group of Pathogens	Pathogens		n (%)
**Gram-positives, 281 (11.2)**		*Enterococcus*		241 (85.8)
			*faecalis*	209
			*faecium*	26
			spp.	6
		*Streptococcus agalactiae*		24 (8.5)
		*Staphylococcus aureus*		16 (5.7)
**Gram-negatives, 2217 (88.8)**	*Enterobacterales*			2116 (95.4)
		*Escherichia coli*		1324
		*Klebsiella pneumoniae*		308
		*Klebsiella oxytoca*		164
		*Enterobacter cloacae*		106
		*Enterobacter aerogenes*		60
		*Citrobacter koseri*		31
		*Proteus mirabilis*		32
		*Citrobacter freundi*		27
		*Serratia marcescens*		22
		Other		42
			*Klebsiella aerogenes*	8
			*Enterobacter* spp.	6
			*Klebsiella* spp.	6
			*Morganella morganii*	4
			*Serratia* spp.	3
			*Citrobacter* spp.	3
			*Citrobacter farmeri*	2
			*Enterobacter hormaechei*	2
			*Salmonella* spp.	2
			*Enterobacter kobei*	1
			*Citrobacter braakii*	1
			*Citrobacter werkmanii*	1
			*Proteus* spp.	1
			*Serratia liquefaciens*	1
			*Serratia plymuthica*	1
	Glucose non fermenting			101 (4.6)
		*Pseudomonas aeruginosa*		90
		Other		11
			*Pseudomonas* spp.	5
			*Stenotrophomonas maltophilia*	2
			*Acinetobacter lwoffii*	1
			*Acinetobacter* spp.	1
			*Pseudomonas fluorescens*	1
			*Pseudomonas putida*	1

**Table 3 microorganisms-13-00607-t003:** Distribution of the most frequently bacterial pathogens isolated (n = 2201) with respect to sex, age (<3 months, ≥3 months), and period (<2015, ≥2015) of the first isolate.

	*Escherichia coli*, n = 1324	*Klebsiella pneumoniae*, n = 308	*Klebsiella oxytoca*, n = 164	*Enterobacter cloacae*, n = 106	*Pseudomonas aeruginosa*, n = 90	*Enterococcus faecalis*, n = 209	*p*-Value
Sex, F, n (%)	496 (37.5)	129 (41.9)	55 (33.5)	36 (34)	38 (42.2)	80 (38.3)	0.424
Sex, M, n (%)	828 (62.5)	179 (58.1)	109 (66.5)	70 (66)	52 (57.8)	129 (61.7)	
Age, <3 months, n (%)	618 (46.7)	193 (62.7)	98 (59.8)	62 (58.5)	36 (40.0)	139 (66.5)	<0.001
Age, ≥3 months, n (%)	706 (53.3)	115 (37.3)	66 (40.2)	44 (41.5)	54 (60.0)	70 (33.5)	
Year of the first UTI, <2015, n (%)	687 (51.9)	146 (47.4)	106 (64.6)	59 (55.7)	57 (63.3)	131 (62.7)	<0.001
Year of the first UTI, ≥2015, n (%)	637 (48.1)	162 (52.6)	58 (35.4)	47 (44.3)	33 (36.7)	78 (37.3)	

**Table 4 microorganisms-13-00607-t004:** Distribution of antibiotic resistance overall and by period (<2015 or ≥2015) of the first isolate.

	Susceptible, n	Resistant, n	Not Tested, n	% Resistant/Tested	% Resistant/Tested <2015	% Resistant/Tested ≥2015	*p*-Value
** *Oral* **							
Amoxicillin-clavulanate							
*Escherichia coli*	937	362	25	27.9	24.1	31.8	0.002
*Klebsiella pneumoniae*	188	115	5	37.9	37	38.8	0.738
*Klebsiella oxytoca*	133	30	1	18.4	21.7	12.3	0.139
*Enterobacter cloacae*	0	99	7	100	100	100	
Cefixime							
*Escherichia coli*	336	27	961	7.4	7.4	-	
*Klebsiella pneumoniae*	69	17	222	19.8	19.8	-	
*Klebsiella oxytoca*	45	9	110	16.7	16.7	-	
*Enterobacter cloacae*	12	18	76	60	60	-	
Cefuroxime							
*Escherichia coli*	590	40	694	6.3	6.3	-	
*Klebsiella pneumoniae*	108	31	169	22.3	22.3	-	
*Klebsiella oxytoca*	77	21	66	21.4	21.4	-	
*Enterobacter cloacae*	3	26	77	89.7	89.7	-	
Ciprofloxacin							
*Escherichia coli*	1216	102	6	7.7	5	10.7	<0.001
*Klebsiella pneumoniae*	278	24	6	7.9	2.7	12.8	0.001
*Klebsiella oxytoca*	163	0	1	0	0	0	
*Enterobacter cloacae*	102	2	2	1.9	1.7	2.2	1.000
*Pseudomonas aeruginosa*	83	3	4	3.5	3.6	3.3	1.000
Fosfomycin							
*Escherichia coli*	1159	8	157	0.7	0.5	0.8	0.728
*Klebsiella pneumoniae*	213	52	43	19.6	22.1	17.8	0.377
*Klebsiella oxytoca*	107	31	26	22.5	23.5	21	0.739
*Enterobacter cloacae*	57	30	19	34.5	26.7	42.9	0.112
Nitrofurantoin							
*Escherichia coli*	1279	3	73	0.2	0.2	-	
*Klebsiella pneumoniae*	59	39	210	39.8	39.8	-	
*Klebsiella oxytoca*	70	5	89	6.7	6.7	-	
*Enterobacter cloacae*	27	12	67	30.8	30.8	-	
*Enterococcus faecalis*	133	3	73	2.2	2.2	-	
Cotrimoxazole							
*Escherichia coli*	846	273	205	24.4	23.9	25.1	0.656
*Klebsiella pneumoniae*	220	46	42	17.3	12.3	23.3	0.018
*Klebsiella oxytoca*	146	2	16	1.3	0.9	2.4	0.488
*Enterobacter cloacae*	83	6	17	6.7	10.2	0	0.093
** *Intravenous* **							
Cefotaxime							
*Escherichia coli*	1050	71	203	6.3	5.7	6.8	0.472
*Klebsiella pneumoniae*	210	47	51	18.3	11.9	22.4	0.033
*Klebsiella oxytoca*	123	4	37	3.1	5.6	0	0.130
*Enterobacter cloacae*	50	34	22	40.5	35.9	44.4	0.426
Ceftazidime							
*Escherichia coli*	1260	59	5	4.5	3.5	5.5	0.075
*Klebsiella pneumoniae*	246	56	6	18.5	13.7	23.1	0.036
*Klebsiella oxytoca*	160	3	1	1.8	2.8	0	0.552
*Enterobacter cloacae*	69	35	2	33.6	27.1	42.2	0.106
*Pseudomonas aeruginosa*	79	7	4	8.1	7.1	10	0.691
Ampicillin							
*Enterococcus faecalis*	195	2	12	1.0	1.5	0	0.549

## Data Availability

The original contributions presented in this study are included in the article. Further inquiries can be directed to the corresponding author.

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
