# Peer review of "Etiology and Oral Antibiotic Susceptibility Patterns of the First Urinary Tract Infection Episode in Infants Under 6 Months of Age: A 17-Year, Retrospective, Single-Center Study in Italy"

_microorganisms, 2025, doi:10.3390/microorganisms13030607_

Round 1
Reviewer 1 Report
Comments and Suggestions for Authors
Francesca Bagnasco et al have coducted a retrospective study aimed to describe the distribution of uropathogens and their antibiotic susceptibility, focusing on oral formulations. They have concluded that Escherichia coli remains the most common uropathogen while Klebsiella pneumoniae has not only shown a high prevalence but also significant resistance, particularly in recent years.
This issue is very interesting as UTI infections is one of the commonest infections during childhood and not free of severe complications. In general the study despite its retrospective nature is written clearly and it is easy to be followed by the reader. The methods are well described and the results are presented clearly while the discussion is relative to the results of the present study.
Comments and suggestions:
In introduction section the authors may want to add a paragraph described the hypothesis and the aim of this study ie we have hypothesized (based on the literature or on our own experience ) that the type or the resistance of the bacteria causing UTI infections have been changed through the years. Thus a study was conducted to answer this hypothesis by….
The 2.8 months period ( from median ) seems somehow arbitrary. On the other hand neonatal and early infancy (up to 3 months of age) is a distinct period regarding the reaction of the neonate or young infant to infection comparing with late infancy .. In our opinion before 3 months and after 3 months would be more physiologically applicable.
The limitations of the study should be gathered to a special paragraph before the section Conclusions.
Author Response
Comments 1: In introduction section the authors may want to add a paragraph described the hypothesis and the aim of this study ie we have hypothesized (based on the literature or on our own experience ) that the type or the resistance of the bacteria causing UTI infections have been changed through the years. Thus a study was conducted to answer this hypothesis by….
Response 1: We thank the Reviewer for the positive feedback and for pointing this out. We agree with this comment. Therefore, a paragraph has been added in the introduction (starting on line 64 in the tracked changes file).
Comments 2: The 2.8 months period (from median) seems somehow arbitrary. On the other hand neonatal and early infancy (up to 3 months of age) is a distinct period regarding the reaction of the neonate or young infant to infection comparing with late infancy .. In our opinion before 3 months and after 3 months would be more physiologically applicable
Response 2: Agree, the cut-of 3 months is more physiologically applicable. We have, accordingly, modified the cut-off of age and the related results (Table 3) in the tracked changes file.
Comments 3 The limitations of the study should be gathered to a special paragraph before the section Conclusions.
Response 3: A limitation section has been added in the discussion before the paragraph of the conclusions (line 310 in the tracked changes file).
Reviewer 2 Report
Comments and Suggestions for Authors
Would suggest giving a definition for age of an infant.
Suggest adding a reference for sentence starting on line 43.
Suggest adding a reference for sentence ending on line 54.
Sentence starting on line 57 refers to increasing ESBL producing bacteria in children. Is there some reference to this for infants that could be used instead?
Considering adding some discussion in the discussion section regarding the use of CLSI guidelines and EUCAST guidleines during different years of this research.
Author Response
Comments 1: Would suggest giving a definition for age of an infant.
Response 1: We thank the Reviewer for the positive feedback and for pointing this out. We agree, "infant" is the term generally used to describe children from 0 to 12 months. Now we have used "young infant," which we believe is more appropriate for defining infants who are less than 6 months old.
Comments 2: Suggest adding a reference for sentence starting on line 43.
Response 2: The reference n.1 has been added in the tracked changes file.
Comments 3: Suggest adding a reference for sentence ending on line 54.
Response 3: The references n.6 and n.15 have been added in the tracked changes file.
Comments 4: Sentence starting on line 57 refers to increasing ESBL producing bacteria in children. Is there some reference to this for infants that could be used instead?
Response 4: The references n.13 and n.14 have been added in the tracked changes file.
Comments 5: Considering adding some discussion in the discussion section regarding the use of CLSI guidelines and EUCAST guidleines during different years of this research.
Response 5: We agree with this comment. Therefore, a paragraph has been added in the discussion (starting on line 211 in the tracked changes file).
